# TDP-43 pathology in *Drosophila* induces glial-cell type specific toxicity that can be ameliorated by knock-down of SF2/SRSF1

**Sarah Krupp[1], Isabel Hubbard[1], Oliver Tam[2¤], Gale M. Hammell[2¤], Josh Dubnau[1,3]***

**1** Program in Neuroscience, Department of Neurobiology and Behavior, Stony Brook University, New York, United States of America, **2** Cold Spring Harbor Laboratory, Cold Spring Harbor, New York, United States of America, **3** Department of Anesthesiology, Stony Brook School of Medicine, New York, United States of America

¤ Current address: Institute for Systems Genetics, Department of Neuroscience & Physiology, NYU Langone Health New York, United States of America

* Joshua.dubnau@stonybrookmedicine.edu

**Data Availability Statement:** All of the raw sequencing data are available on GEO (GSE242209;https://www.ncbi.nlm.nih.gov/geo/query/acc.cgi?acc=GSE242209) All other data are

## Abstract

Accumulation of cytoplasmic inclusions of TAR-DNA binding protein 43 (TDP-43) is seen in both neurons and glia in a range of neurodegenerative disorders, including amyotrophic lateral sclerosis (ALS), frontotemporal dementia (FTD) and Alzheimer's disease (AD). Disease progression involves non-cell autonomous interactions among multiple cell types, including neurons, microglia and astrocytes. We investigated the effects in *Drosophila* of inducible, glial cell type-specific TDP-43 overexpression, a model that causes TDP-43 protein pathology including loss of nuclear TDP-43 and accumulation of cytoplasmic inclusions. We report that TDP-43 pathology in *Drosophila* is sufficient to cause progressive loss of each of the 5 glial sub-types. But the effects on organismal survival were most pronounced when TDP-43 pathology was induced in the perineural glia (PNG) or astrocytes. In the case of PNG, this effect is not attributable to loss of the glial population, because ablation of these glia by expression of pro-apoptotic reaper expression has relatively little impact on survival. To uncover underlying mechanisms, we used cell-type-specific nuclear RNA sequencing to characterize the transcriptional changes induced by pathological TDP-43 expression. We identified numerous glial cell-type specific transcriptional changes. Notably, SF2/SRSF1 levels were found to be decreased in both PNG and in astrocytes. We found that further knockdown of *SF2/SRSF1* in either PNG or astrocytes lessens the detrimental effects of TDP-43 pathology on lifespan, but extends survival of the glial cells. Thus TDP-43 pathology in astrocytes or PNG causes systemic effects that shorten lifespan and *SF2/SRSF1* knockdown rescues the loss of these glia, and also reduces their systemic toxicity to the organism.

included in the main figures and tables and in the supplemental figures and tables.

**Funding:** This work was supported by grants RF1AG057338, R01AG078788 and RF1AG076493 from the NIA to JD. The funders had no role in study design, data collection and analysis, decision to publish, or preparation of the manuscript.

**Competing interests:** The authors have declared that no competing interests exist.

## Author summary

Neurodegenerative disorders such as amyotrophic lateral sclerosis (ALS), frontotemporal dementia (FTD) and Alzheimer's disease (AD) involve dysfunction in both neuronal and non-neuronal cells in the brain. Each of these disorders involves abnormal accumulation of a protein called TDP-43, which is thought to cause toxic effects that ultimately lead to cell death. We used cell type specific sequencing to characterize the effects of inducing pathological TDP-43 within each of the non-neuronal cell types in Drosophila. This identified changes in expression of a gene called SF2/SRSF1, which has previously been implicated in mediating toxic effects. We found that knocking down the expression of this gene within individual glial cell types was sufficient to prevent the loss of nearby neurons.

## Introduction

Pathological cytoplasmic inclusions of TDP-43 are seen in post-mortem brain tissues from >90% of patients with ALS, ~45% of FTD cases, and nearly 50% of AD cases [1–8]. TDP-43 is an RNA and DNA binding protein that is normally found in the nucleus of healthy cells, where it performs a variety of cellular functions, including regulation of splicing, transcription, and silencing of retrotransposons (RTEs) and endogenous retroviruses (ERVs) [5,9–13]. Although a few percent of familial ALS cases are caused by amino acid substitutions in the TDP-43 protein coding region, the vast majority of cases involve inclusions with TDP-43 protein of normal amino acid sequence [3,14]. The upstream mechanisms that trigger TDP-43 aggregation are poorly understood, but TDP-43 protein levels are under tight regulatory control, including via an autoregulatory mechanism in which nuclear TDP-43 binds to its own transcript to inhibit protein expression [15].

Under pathological conditions, TDP-43 is lost from the nucleus and becomes sequestered into cytoplasmic inclusions, which removes this autoregulatory mechanism [15–17]. This may further increase cytoplasmic concentrations of the protein. The most commonly used animal models rely on overexpression of transgenic TDP-43 to drive aggregation [16–19]. Such overexpression largely sidesteps the role of upstream disease initiators. Although it is an imperfect representation of the spontaneous appearance of pathology in sporadic disease, overexpression can drive robust accumulation of hyperphosphorylated, insoluble cytoplasmic inclusions and loss of nuclear localization [3,5,20].

The loss of nuclear function is in fact a shared feature of over-expression and knock out models, but the over-expression approach also has the potential to model impacts from presence of cytoplasmic inclusions [21]. Over-expression models of TDP-43 pathology have revealed myriad cellular roles for TDP-43 as well as downstream effects of pathological TDP-43 that are found to be relevant to disease [5,21–25]. The majority of animal model studies have focused on impacts of TDP-43 pathology in neurons. In this study, we investigated the mechanisms at play in glia, including specialized glial cell types that form the Drosophila blood brain barrier (BBB). There are several motivations for these experiments. First, pathological TDP-43 in patients is observed both in neurons and in glial cells [2,24–29] and there are established roles in disease progression of both astrocytes and microglia [30–34]. For example, primary astrocytes derived from post-mortem sporadic ALS patients are toxic to motor neurons as revealed either with co-culture or application of astrocyte conditioned media [35,36].

Such astrocyte toxicity has also been observed in in vivo rodent and fly models [35–37]. Activated microglia also may stimulate astrocytes to adopt a pro-inflammatory state that drives

non-cell-autonomous toxicity [34,38]. In addition to these glial impacts, there are clear effects of blood brain barrier dysfunction [39–41]. Findings in a *Drosophila* TDP-43 model are convergent with this literature, implicating a variety of non-neuronal cells in toxicity to neurons [37,42]. We have previously demonstrated that initiating TDP-43 protein pathology by overexpression in *Drosophila* glial cells is sufficient to cause severe motor deficits, drive TDP-43 pathology in nearby neurons, cause neuronal cell death, and greatly reduce lifespan [20,37,42].

*Drosophila* have five major glial cell types in the adult central brain, whose functional roles and anatomical features bear many similarities to non-neuronal cells (including glia) in the mammalian brain [43,44]. The perineurial glia (PNG) and subperineurial glia (SPG) form an epithelial like BBB on the brain surface, held together by cell-cell junctions [45]. Although they have a glial origin, the SPG undergo a mesenchymal-to-epithelial transition [43]. In addition to these specialized BBB forming glial cells, the cortex glia (CG), astrocyte-like glia (ALG), and ensheathing glia (EG) provide many of the functional characteristics of mammalian astrocytes and microglia [43–47]. We previously reported that TDP-43 pathology in astrocytes, PNG and SPG each contribute non-cell autonomous effects to nearby neurons [37,42].

Here, we used cell-type-specific nuclear RNA-seq both to better characterize the baseline transcriptional profiles of the 5 major glial cell types and to investigate mechanisms underlying glial cell type specific impacts of TDP-43 pathology. We identified numerous glial cell-type-specific transcriptional changes in response to induced TDP-43 expression. Notably, these included *SF2/SRSF1*, a known suppressor of *C9orf72* ALS/FTD models [48] that we also recently identified in a screen for suppression of TDP-43 toxicity in motor neurons [49]. We observe that in response to TDP-43 pathology, SF2/SRSF1 is down-regulated in both the PNG and SPG specialized BBB glia. We demonstrate among the 5 glial cell types, TDP-43 toxicity in PNG is a primary driver of organismal toxicity. And we demonstrate that the systemic effects on the animal's survival from TDP-43 pathology in PNG can be ameliorated by PNG specific knockdown of *SF2/SRSF1*.

## Results

### Organismal impacts from glial cell-type specific TDP-43 pathology reveals PNG as a primary driver of systemic toxicity

We have previously demonstrated that in *Drosophila*, pan-glial over-expression of TDP-43 causes loss of nuclear localization and accumulation of cytoplasmic hyperphosphorylated TDP-43 [20,42]. Such pan-glial TDP-43 toxicity reduces lifespan and causes non-cell autonomous toxicity to neurons that appears to involve some combination of astrocytes, PNG and SPG [37,42]. Importantly, the toxicity to neurons from glial-specific TDP-43 over-expression in a humanized *Drosophila* model also triggers pathological cytoplasmic accumulation of TDP-43 in nearby neurons where TDP-43 is not over-expressed [42]. We sought to distinguish the contribution of TDP-43 proteinopathy in each glial cell type to the animal's survival. Using cell-type specific Gal4 lines (methods) and the temperature sensitive Gal80 (Gal-80$^{ts}$) approach, we induced post development over-expression of TDP-43 in each of the 5 individual glial cell types. This temperature switch expression system allowed us to avoid effects on development from over-expressing TDP-43 during glial specification, and also permitted us to determine the time-course of lethality after induction on the first day of adult life (Fig 1A). In order to monitor effects on survival of each glial cell type, we co-expressed a nuclear-envelope tagged fluorescent reporter (5X-UAS-unc84-2X-GFP) [37,42]. We then monitored the effects over time of inducing glial-cell type specific TDP-43 over-expression on both organismal and cell type survival. We found induced TDP-43 expression in each of the five glial cell types was sufficient to significantly reduce lifespan (Fig 1B–1F) though we note this reduction is quite

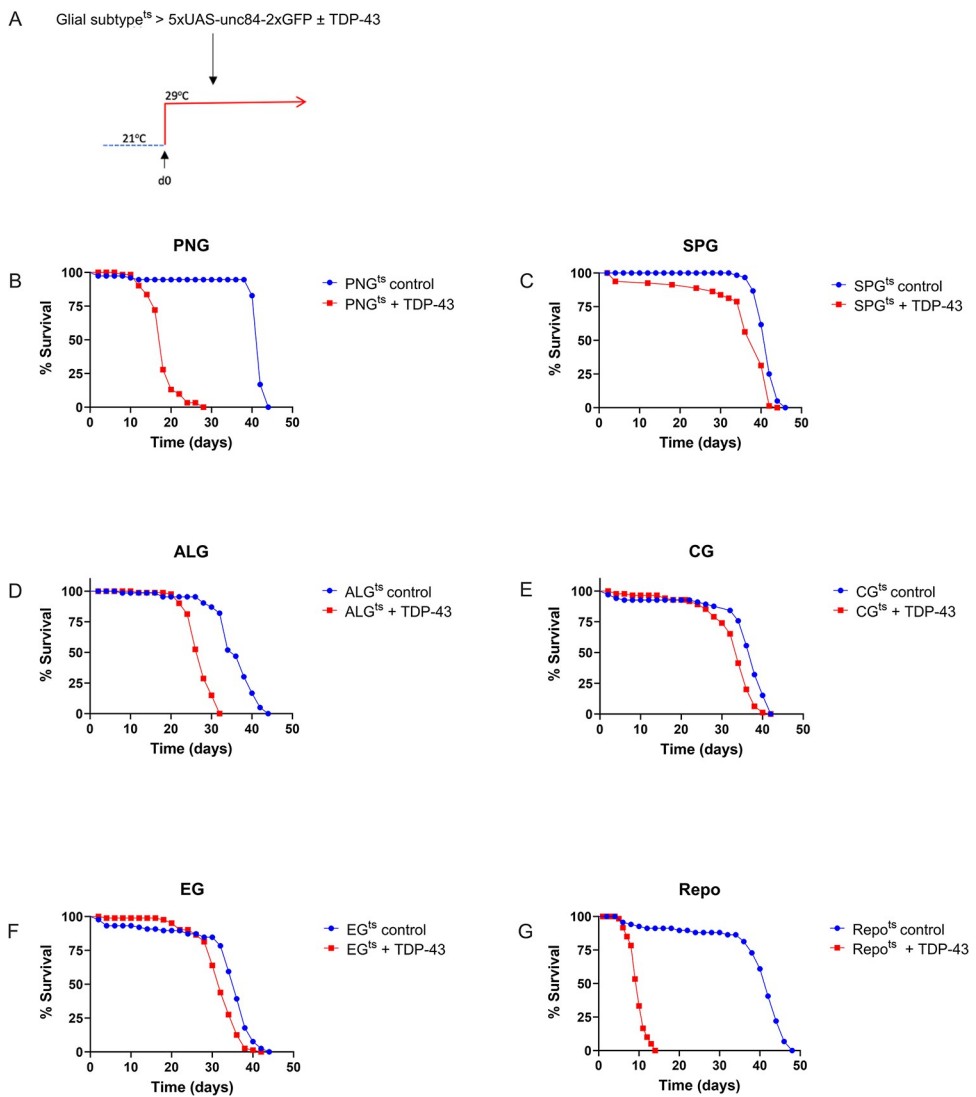

**Fig 1. Effect on lifespan of post-developmental TDP-43 overexpression in individual glial cell types of the *Drosophila* CNS.** (A) To induce post development expression of hTDP-43 and 5XUAS-Unc84-2XGFP, flies were reared at 21˚C and shifted to 29˚C upon eclosion (day 0). (B-F) Lifespan analyses with expression in each individual glial cell type, compared to a control without TDP-43; Effects were significant (Log-Rank) for each glial cell type (PNG, P< 9.003e-28; SPG, P< 1.596e-7; ALG, P< 9.198e-24; CG, P< 0.00000197; EG, P< 0.000004475). (G) Lifespan analysis for pan-glial TDP-43 expression compared to a control without TDP-43 (Repo) P<2.231e-27. Gal4 lines and full genotypes used for each cell type are listed in methods.

modest in size for the cases of the EG and CG and is intermediate in magnitude for the case of SPG. In contrast, we see more substantial reduction in lifespan for PNG (median survival reduced from 42 for control to 18 days) and ALG (median survival reduced from 36 vs. 28 days).

It is challenging to make direct comparisons between the severity of lethality with TDP-43 over-expression in each of the 5 glial cell types because the levels of expression may vary across cell types. But it is noteworthy that the effects on lifespan with induced TDP-43 pathology in PNG (Fig 1B) were particularly severe, and approached the magnitude of effect on lifespan seen with pan-glial hTDP-43 expression (Fig 1G). Thus, inducing TDP-43 pathology within

each of the 5 glial cell types does impact survival of the animals, but the magnitude of effect on lifespan from TDP-43 pathology is most severe with PNG.

## Similar intracellular toxicity of induced TDP-43 across glial cell types

The observed differences in magnitude of effects on lifespan seen above could result from (1) differential intracellular sensitivity of a given cell type to TDP-43 toxicity, (2) differences in severity of effects on brain homeostasis from loss of each glial cell type, or (3) from cell-type specificity in non-cell autonomous toxic effects from TDP-43 proteinopathy [37,42]. We first tested whether there were differences in sensitivity to the toxicity of induced TDP-43 proteinopathy within each glial cell type. This could derive from either biological differences in the resistance of each cell type, from differences in expression levels within that cell type, or from a combination of the two. To distinguish these possibilities, we took advantage of the co-induced 5X-UAS-unc84-2X-GFP fluorescent reporter, to monitor cell number over a time-course after induction of TDP-43. With each genotype, we dissected brains at 2, 5, and 10-days post-induction of both TDP-43 and used confocal microscopy to quantify the numbers of nuclei labeled with GFP.

With each cell type, we observed a progressive loss in the numbers of labeled nuclei over time. With PNG, ALG, EG and CG, we see a similar time-course of cell loss (Fig 2). With each of those four glial cell types, there is a near complete loss of labeled nuclei by 10 days post induction. With SPG, the rate of loss appears to be somewhat slower (2C and 2D). Despite the fact that PNG, ALG, EG and CG appear to be lost at similar rates, TDP-43 over-expression in PNG and ALG caused a palpably more severe shortening in lifespan compared to the other cell types, with the PNG expression causing the most severe effect (Fig 1). The SPG exhibit an intermediate severity of lifespan impact and a slower loss of cell number, making comparisons more difficult. But comparing PNG with EG and CG, there is a starkly higher organismal impact that cannot be explained by differences in rate of loss of these glial subtypes.

## Lifespan deficit observed with PNG cell-type-specific TDP-43 induction is not recapitulated by PNG ablation

Given the severe impact on lifespan that we observe with PNG specific TDP-43 induction, we wondered if this reflected loss of an essential function that is specific to this specialized group of BBB glial cells. To test this idea, we compared the effects on lifespan of inducing expression in PNG of TDP-43 vs the pro-apoptotic gene reaper (*rpr*). In order to quantify effects on PNG cell loss, we co-expressed a UAS-nuclear-GFP. In each case, we used the same Gal-80$^{ts}$ approach to induce post development expression of GFP and either TDP-43 or rpr. As above, animals were reared at 21˚C and then shifted to 29˚C after eclosion (Day 0) in order to induce expression. We then quantified the effects on lifespan and PNG cell loss (Fig 3).

As with induction of TDP-43 pathology, expression of rpr causes progressive loss of the PNG cell population. This occurs with similar kinetics across both groups (Fig 3A and 3B). Despite the indistinguishable effects on PNG cell survival, we find that lifespan of the animals after inducing TDP-43 pathology in PNG is greatly reduced compared to lifespan after induction of *rpr* (Fig 3C, p<0.0001). Thus, the lifespan deficit resulting from TDP-43 pathology in PNG cannot be explained merely by a loss of this population of glial cells. Instead, this supports the interpretation that TDP-43 pathology in PNG causes systemic effects that reduce survival[42] of the animals.

## Glial-cell type specific RNA-sequencing reveals differential effects of TDP-43

In order to investigate glial cell-type specific impacts, we profiled both baseline expression in each glial cell type, and differential expression in response to TDP-43 induction. To

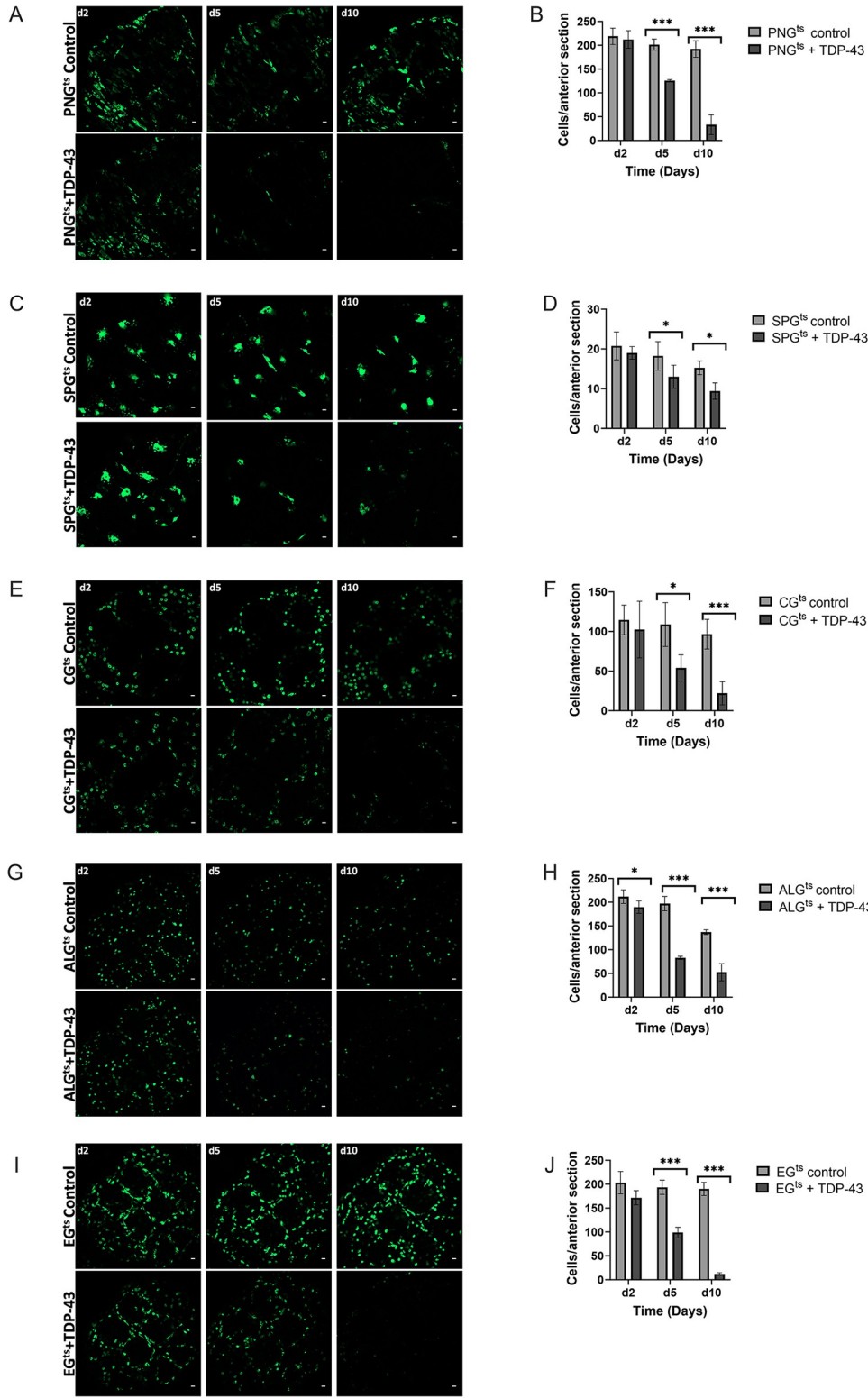

**Fig 2. TDP-43 overexpression leads to age-dependent loss of each glial cell type.** Representative confocal images (Scale bar = 10μm.) of equivalent anterior section of the *Drosophila* brain on days 2,5, and 10 post eclosion for flies control (top) vs flies that express TDP-43 in PNG^ts (A), SPG^ts (C), CG (E) ALG (G), EG (I). Quantification of the number of each glial cell type in an equivalent anterior section was determined by counting nuclei labeled by Unc84-2X-GFP, which was co-expressed (B,D,F,H,J). Mean and SD are shown. Students T-test used to compare genotypes

within each timepoint. * p<0.05, ***p<0.001. Gal4 lines and full genotypes used for each cell type are listed in methods.

accomplish this, we used RNA sequencing of nuclei purified from individual glial cell types using tandem affinity purification of INTACT nuclei (TAPIN) [50,51]. Here too, we used the Gal80ts approach to induce the UAS-unc84-2X-GFP nuclear tag either alone or with TDP-43.

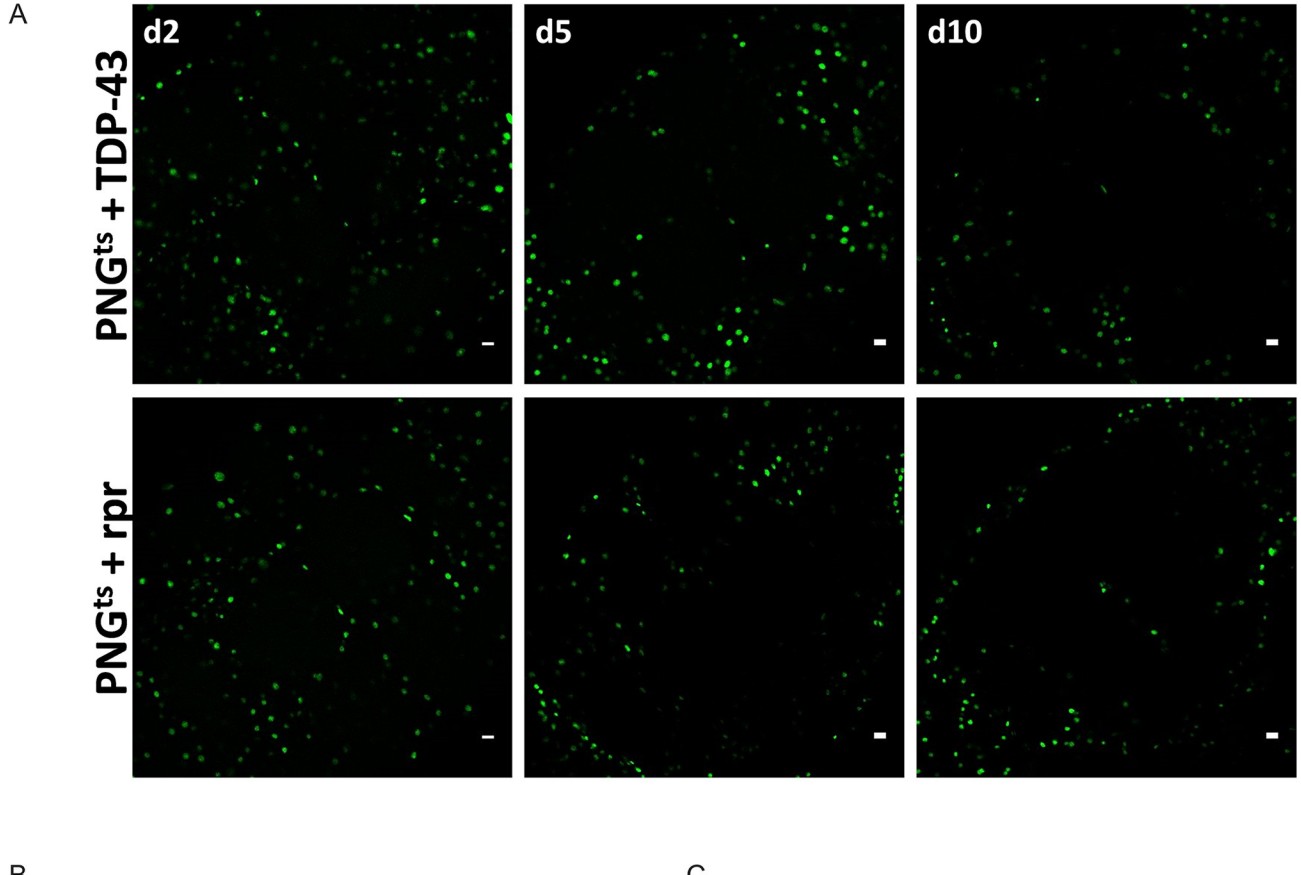

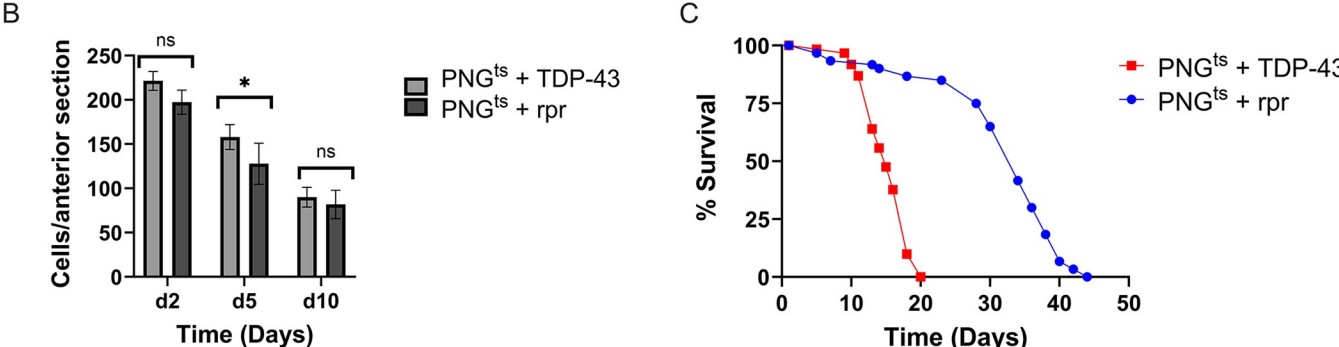

**Fig 3. Lifespan deficit observed with TDP-43 expression in PNG is not recapitulated by PNG ablation.** (A) confocal images (Scale bar = 10μm) of an anterior section of the *Drosophila* brain on days 2,5, and 10 for flies expressing of PNG<sup>ts</sup> + TDP-43 (top) or PNG<sup>ts</sup> + rpr (bottom). (B) quantification of PNG cells in equivalent anterior sections was performed on days 2,5, and 10 post induction by counting the number of nuclei labeled with GFP, which was co-expressed using the UAS-WM reporter (Methods). Mean and SD shown for each. * p<0.05, T-test. (C) Lifespan analysis of flies expressing rpr (PNG<sup>ts</sup> + rpr) or TDP-43 (PNG<sup>ts</sup> + TDP-43). TDP-43 expression in PNG yields a significantly shorter lifespan than does rpr expression (Log rank, P<1.085e-20). Gal4 lines and full genotypes are listed in methods.

**Table 1. Nuclear RNA-seq from each of five glial cell types, with and without TDP-43 induction.** TAPIN RNA-seq library metrics at day 2 across each glial cell type with and without TDP-43.

| Library Metric | PNGts+TDP43/PNGts Control (N = 3) | SPGts+TDP43/SPGts Control (N = 3) | CGts+TDP43/CGts Control (N = 3) | ALGts+TDP43/ALGts Control (N = 2/3) | EGts+TDP43/EGts Control (N = 3) |
|---|---|---|---|---|---|
| Total# of reads | 70,421,315/62,494,049 | 81,368,497/77,664,506 | 64,108,924/103,515,309 | 48,552,166/62,506,607 | 71,281,357/74,018,004 |
| % mapped (avg) | 86.7/85.9 | 83.4/84.5 | 84.8/84.7 | 82.3/84.2 | 85.2/85.0 |
| % iniquely aligned (avg) | 53.3/58.2 | 52.7/54.7 | 48.8/64.3 | 65.4/70.0 | 58.7/65.9 |
| Features deteted | 15 315 | 15 009 | 14 301 | 14 354 | 14 954 |
| # of upreg features (P<0.05 | 1 289 | 1 012 | 2 931 | 1 483 | 2 498 |
| # of downreg features (P<0.05) | 748 | 782 | 2 316 | 1 025 | 1 772 |

The 5X-UAS-unc84-2X-GFP reporter is tethered to the inner nuclear membrane, but has the GFP molecule on the outer surface of the nucleus. This provides the means for affinity purification of labeled nuclei from heterogeneous tissue (S1 Fig, S1 Table). In this TAPIN method, GFP-expressing nuclei are purified via two rounds of affinity purification, and then nuclear RNA is extracted and used to produce barcoded Illumina sequencing libraries.

Using this method, we generated nuclear RNA sequencing libraries from each of the 5 cell types and compared expression 2 days after induction of either the GFP tag alone (baseline), or the GFP tag plus TDP-43 (N = 3 libraries per group). Because each of the cell types are comprised of different numbers of cells per brain (S1 Table), we adjusted the number of brains to yield approximately similar numbers of cells per library. Of these 30 samples, 28 yielded high read depth and passed quality control metrics (Table 1, Methods). One CG baseline library and one ALG baseline library were removed from analysis due to low sequencing depth (<5 million reads for one CG sample) and poor correlation with other samples (for one ALG sample). Among the remaining 28 libraries the total number of reads ranged from ~62.5 million to ~103.5 million, with an average of 85% of reads mapped to the reference genome (Table 1). This corresponded to detection of ~15,000 features [i.e. genes, transposable elements (TEs), structural RNAs] on average.

When comparing the baseline expression profiles across each of the five glial cell types, several things are of note. First, we found that the CG and EG show higher correlation in gene expression with each other than with the other cell types (S2A and S2B Fig). Similarly, the PNG and SPG, which constitute the BBB, are more similar to each other in gene expression profiles than they are with the other cell types (S2B and S2C Fig). Within each cell type, we also found that induction of TDP-43 over-expression resulted in differential expression in numerous genes, including ones that were cell-type specific. Overall, differential expression analysis with DESeq2 revealed upregulation of ~1,000 to ~3,000 features and downregulation of ~750 to ~2,300 features across the various cell types. The majority of the differentially expressed features were cellular genes (Fig 4A,4C,4E,4G and 4I, S2 Table), although we also detected many changes in expression of TEs, including RTEs and ERVs as expected (Fig 4B,4D,4F,4H and 4J; S3 Table) [20,52–57].

The transcriptional responses to TDP-43 pathology within each glial cell type included significant differential expression of many genes from cellular pathways that previously have been implicated in TDP-43 related neurodegeneration. These included genes involved in neuroinflammation [52,53], DNA damage/repair [54–56], chromatin organization [49,57] and nucleocytoplasmic transport [58], and many transposable elements [20,52–57], each of which have been extensively implicated in TDP-43 related neurodegeneration (S3–S7 Table).

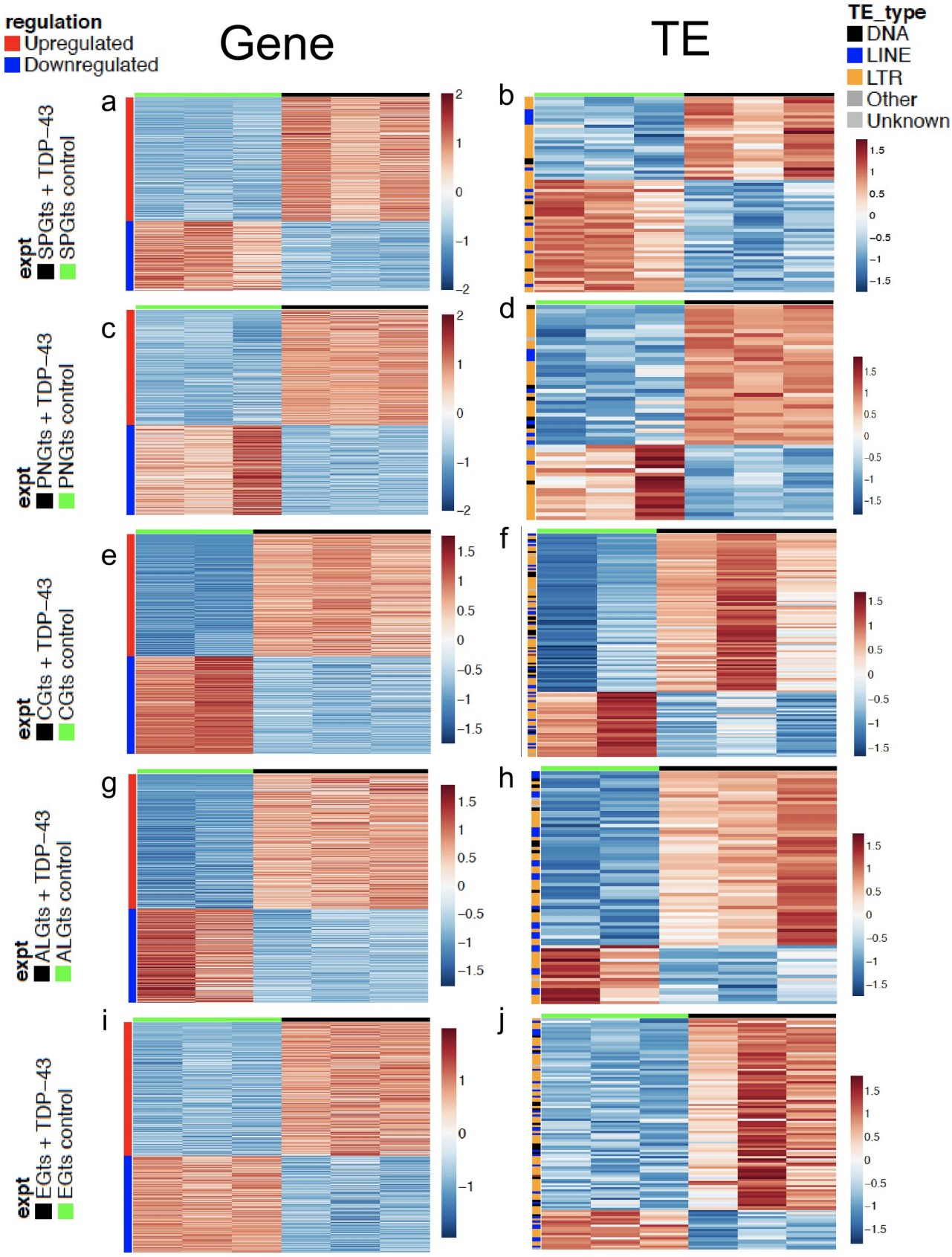

**Fig 4. Differentially expressed genes and TEs in response to TDP-43 overexpression in each glial cell type.** Significantly (p < .05, see methods) upregulated (Red) and downregulated (Blue) genes (A,C,E,G,I) and TEs (B,D,F,H,J) are shown for TDP43 expressing (Green bar, left) vs control (Black bar, right) for each cell type. Genes or TEs that are significantly (p < .05) upregulated and downregulated are shown for PNG (A, B), SPG (C,D), CG (E,F), ALG (G,H) or and EG (I,J). Full genotypes and Gal4 lines are listed in methods.

Although GO analyses do not identify enrichment for effects within these pathways, this may reflect differences in cell-type-specific transcriptomes and/or the fact that we profiled an early timepoint (Day 2 post TDP43 induction), in order to identify early effects. By contrast, expression profiling from human brain samples are at a disease endpoint. Given how soon after induction our expression profiling was done, it was of note that we identified many statistically significant changes in expression of TEs (including RTEs and ERVs), which was true across cell types. Indeed, we found a total of 299 upregulated TEs and 131 downregulated TEs across all five cell types (Fig 4B,4D,4F,4H and 4J; S3 Table). This is convergent with an emerging literature demonstrating that RTEs/ERVs become expressed in response to TDP-43 proteinopathy and may contribute to ALS progression [20,52–57]. In this fly model, we have previously demonstrated that RTEs/ERVs play a key role in mediating the toxicity of TDP-43 in glia, as well as in the intercellular toxicity to neurons [20,37,42].

## Knockdown of SF2/SRSF1 in astrocytes or PNG ameliorates the toxicity of TDP-43

The differential expression analysis identified numerous cell type specific responses to TDP-43, as well as responses that were found to be in common across several cell types (S2C Fig). Among those that were cell type specific, we made particular note of the differential expression of SF2/SRSF1 (S8 Table) because we and others have recently identified knock-down of this gene as a potent suppressor of TDP-43 and C9orf72 pathology [48,49,58]. We previously reported that knockdown of SF2/SRSF1 in animals that overexpress TDP-43 in a subset of motor neurons is sufficient to largely prevent the age-dependent decline in escape locomotion behavior [49]. We also reported that SF2/SRSF1 knock down in glia was sufficient to extend lifespan of animals in which we had induced glial TDP-43 pathology. In that study, we did not examine cell type specificity, but instead used pan-glial manipulations that included all five of the major glial cell types. Our differential expression analysis identified SF2/SRSF1 as being down regulated in response to TDP-43 expression in PNG (-1.31 log2 fold change, p<0.0001) and SPG (-0.85 log2 fold change, p < .001), but not in CG, ALG, or EG (S8 Table).

The fact that SF2/SRSF1 levels are reduced in SPG and PNG, but not other glial cell types, is intriguing given that pan-glial knockdown of SF2/SRSF1 can ameliorate the lifespan effects of pan-glial TDP-43 pathology (Fig 5A) [55]. We observe a substantial impact on lifespan from pathological TDP-43 induction in PNG and ALG (Fig 1), but only see down regulation of SF2/SRSF1 in PNG (S8 Table). We therefore wondered whether cell-type specific knockdown of SF2/SRSF1 would impact the lifespan and cell survival from TDP-43 induction in PNG vs Astrocytes. We used the PNG or Astrocyte Gal4 lines, together with the Gal80[ts] method, to induce TDP-43 together with either a control shRNA against mCherry or the shRNA that targets the SF2/SRSF1 gene[55]. We used the same Gal80[ts] induction conditions described in previous experiments to turn on expression after development (reared at 21˚C, shifted to 29˚C on adult day 0). We found that SF2/SRSF1 knockdown significantly suppressed the lifespan defects caused by TDP-43 induction in either PNG or ALG (Fig 5B and 5C). Thus, despite the fact that TDP-43 induction itself causes down-regulation of SF2/SRSF1 in PNG, further knockdown in PNG has a substantial protective effect that extends organismal survival. And although TDP-43 induction in astrocytes does not on its own cause a decrease in SF2/SRSF1

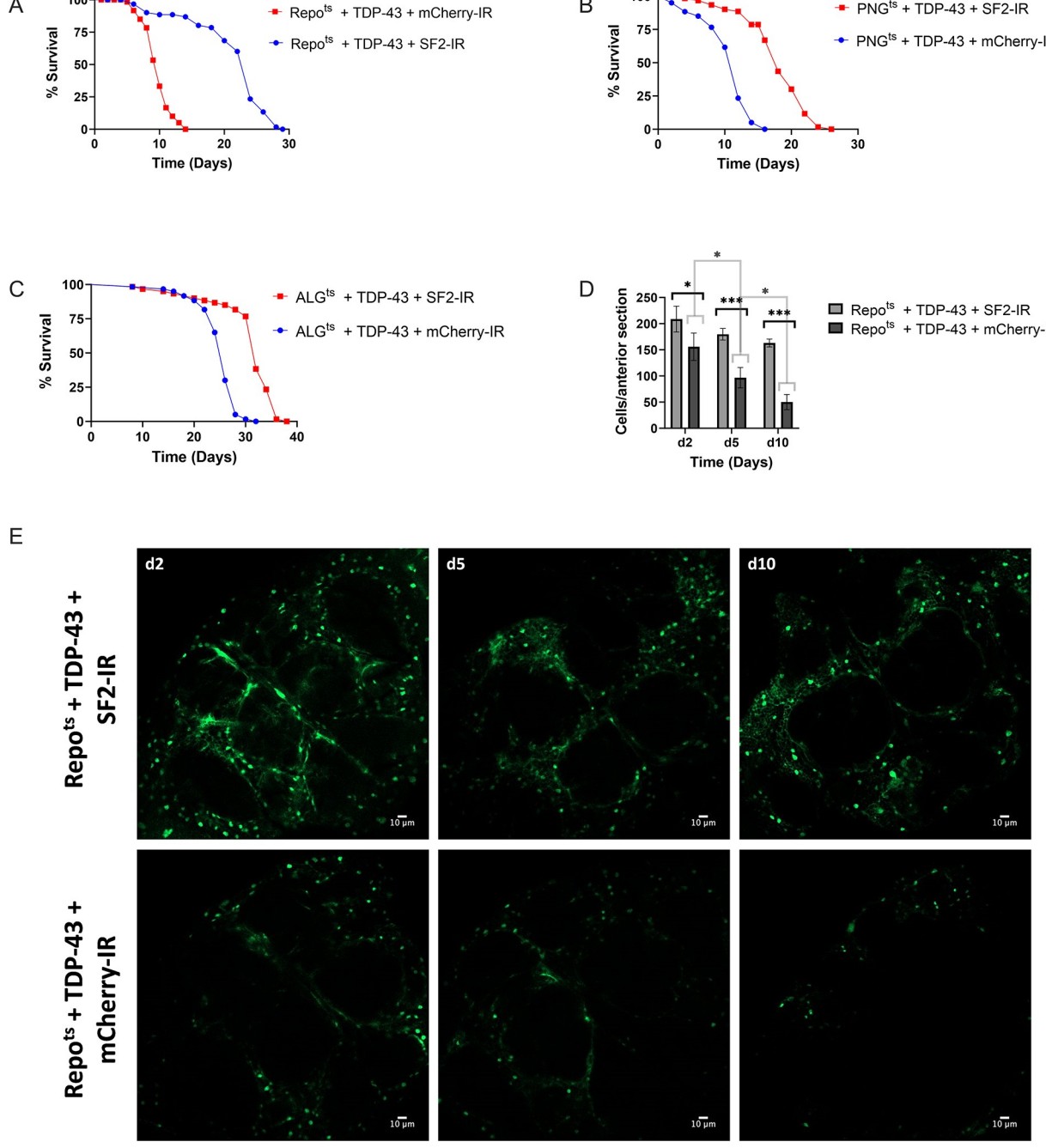

**Fig 5. *SF2/SRSF1* knockdown prevents glial cell loss and extends lifespan after induced TDP-43 overexpression.** (A-C) Lifespan analysis of (A) Repo^ts + TDP-43 + SF2-IR vs Repo^ts + TDP-43 + mCherry-IR, (B) PNG^ts + TDP-43 + SF2-IR vs PNG^ts + TDP-43 + mCherry-IR and (C) ALG^ts + TDP-43 + SF2-IR vs ALG^ts + TDP-43 + mCherry-IR. For each case, SF2/SRSF1 knock down provides a significant extension of lifespan (Log rank test. Repo, P<7.193e-19; PNG, P<1.062e-16; ALG, P<1.081e-22). (D) confocal images (Scale bar = 10μm) of an anterior section of the *Drosophila* brain on days 2,5, and 10 for flies expressing of Repo^ts + TDP-43 + SF2-IR (top) or Repo^ts + TDP-43 + mCherry-IR (bottom). (E) quantification of Repo+ glial cells on days 2,5, and 10 per anterior brain section. Mean and SD are shown. T-tests, * p<0.05, ***p<0.001. Full genotypes and Gal4 lines used are listed in methods.

expression, knock-down in astrocytes of SF2/SRSF1 by RNAi does ameliorate the effects on lifespan from astrocytic induction of TDP-43 pathology.

Inducing TDP-43 in PNG causes loss of most of this population of specialized BBB glia within about 10 days (Fig 2), but the effects on lifespan (Fig 1) are not explained by loss of essential functions of PNG because ablating these cells with rpr expression (Fig 2) does not recapitulate the severe lifespan defect seen with TDP-43 induction. This is consistent with the interpretation that in the presence of TDP-43 pathology, these glia become neurotoxic before they die. Indeed, we previously demonstrated that TDP-43 induction in either SPG, PNG or astrocytes results in non-cell autonomous toxicity that causes the appearance of TDP-43 pathology and DNA damage in neurons, leading to neuronal cell death [37,42]. And in that context, blocking apoptosis of glia was sufficient to prevent glial loss, but this increased toxicity to neurons [37]. In light of these findings, two possibilities could explain the amelioration of the systemic effects on lifespan from knocking down SF2/SRSF1 in PNG or astrocytes. First, it is possible that knock down of SF2/SRSF1 in glia prevents release of a toxic factor that kills neurons. Alternatively, knocking down SF2/SRSF1 may cause glia to die more quickly, thereby preventing their release of a toxic substance. But this second hypothesis can be ruled out because we find that knockdown of SF2/SRSF1 actually delays the loss of glia in response to TDP-43 induction (Fig 5D and 5E). Taken together, these findings support the hypothesis that the protective effects from SF2/SRSF1 are mediated by both an amelioration of the lethality of TDP-43 pathology to these glial cell types, as well as a reduction in the non-cell autonomous toxicity of the glia to surrounding neurons.

## Discussion

We use glial cell-type-specific TDP-43 overexpression in *Drosophila*, which triggers pathological cytoplasmic inclusions, to identify the relative contribution of each cell type to systemic effects on organismal lifespan, cell autonomous effects on glial cell loss, and transcriptional changes within each cell type. The PNG, which constitute a specialized glial cell type that contributes to BBB function [43,44,59], emerged as a primary driver of the effects of TDP-43 induction on lifespan despite similar kinetics of TDP-43 induced cellular loss of each of the glial cell types. It is unlikely that this dramatic effect on lifespan from PNG driven TDP-43 pathology could be explained by loss of essential functions from these specialized BBB cells because ablation of PNG by induced expression of the pro-apoptotic *rpr* gene, does not phenocopy the shortened lifespan seen with TDP-43 expression despite a similar rate of loss of the PNG cell population.

The PNG and SPG, the 'surface glia', are thought to function analogously to the vertebrate BBB. PNG, which comprise the outer most layer of surface glia, far outnumber SPG (~2246 PNG vs ~300 SPG per brain) [43], and lie directly in contact with the hemolymph (analog of blood). Together the PNG and SPG isolate the CNS from hemolymph components by blocking paracellular diffusion [43,45,59,60]. Although BBB function is vital for many aspects of normal physiology [60,61], we demonstrate that loss of these BBB glia is not sufficient to explain the lifespan defects seen when TDP-43 pathology is driven in PNG. Our data argue rather that induced expression of TDP-43 in these glia triggers systemic effects with more severe impact on survival than is seen with ablation of the cell type. This is consistent with the hypothesis that PNG are induced to release toxic factors that contribute to systemic effects that cause shortened lifespan. And our findings also indicate that SF2/SRSF1 plays a key role in mediating the cell autonomous toxicity from TDP-43 over-expression to PNG and astrocytes, but also the systemic toxicity to the animal from such cell-type specific expression.

A sizeable literature supports the idea that release of factors from disease-activated glial cells is sufficient to promote toxicity to other cells including neurons [31,31,62]. This

phenomenology is seen across a variety of models, including vertebrate systems and ALS patient-derived glial cells. In mammalian systems, there is evidence for signaling from neurons to microglia, from microglia to astrocytes, and from astrocytes to neurons. In response to TDP-43 pathology, both microglia and astrocytes are thought to enter activation states that are toxic to neurons. Although *Drosophila* do not have a clear microglial cell type, they do have an astrocyte like cell type. And expression of pathological TDP-43 in Drosophila astrocytes drives non-cell autonomous toxicity to neurons [37]. The findings reported here highlight the impact the PNG, which provide the outermost layer of the BBB. PNG are in direct contact with the hemolymph as well as with the underlying SPG. Thus, the systemic effects of induced TDP-43 within the PNG could involve release of toxic factors into hemolymph. Dysfunction of the BBB from loss of PNG and increased permeability could also synergize with other systemic effects driven by TDP-43 proteinopathy. BBB defects have been noted in patients across a range of neurodegenerative diseases including ALS, FTD and AD. On the other hand, PNG are also in close contact with the underlying SPG, which in turn are in contact with neuronal somata that lie near the surface of the brain.

Our prior work has demonstrated that TDP-43 proteinopathy in *Drosophila* SPG, PNG and astrocytes is sufficient to trigger loss of nearby neurons. Although mechanisms that mediate non-cell autonomous toxic effects in TDP-43 related and other neurodegenerative disorders are not fully understood, there is evidence implicating inflammatory effects [63], movement of pathological aggregates which seed prion-like propagation [64–66] and effects of endogenous retroviruses (ERVs) [42,67]. We have previously shown in the context of this *Drosophila* model that TDP-43 pathology drives expression of the mdg4-ERV, and that mdg4 contributes to the non-cell autonomous effects on neuronal survival [37,42]. Indeed, there is evidence that intercellular movement of ERV generated viral particles may underlie the propagation of toxicity from cell-to-cell [67]. But it is not at all clear whether different mechanisms underlie inter-cellular signaling effects among each glial and neuronal cell type.

The cell-type specific expression profiling reported here provides some insight into the functional differences between glial cell types in *Drosophila*, and potentially to the non-cell autonomous effects of TDP-43. Although several prior publications have reported baseline expression profiling of several glial-cell-types during larval development, and adult stages in *Drosophila*, this is the first direct comparison of RNA expression profiles across all five major glial subtypes in the adult, using a single platform. This dataset supports the idea that CG and EG are functionally most similar, and that PNG and SPG also are more similar to each other than to the other glial cell types (S2 Fig, S2 Table). We anticipate this dataset will serve as a valuable resource to investigate functional properties of each of these glial cell types in *Drosophila*. Differential gene expression, in response to induction of TDP-43 over-expression within each glial cell type identifies a number of genes and signaling pathways that have previously been implicated in neurodegeneration, with some core similarities across cell types, but also many cell type specific effects (Fig 4, S2–S7 Tables). This included numerous differentially expressed TEs (Fig 4 and S3 Table), consistent with previous reports [20,52,68–72]. Notably, the SF2/SRSF1 gene exhibits decreased expression in PNG and SPG, but not other glial cell types (S8 Table).

Several previous reports identified SF2/SRSF1 as a suppressor of both TDP-43 and C9orf72 pathology [48,49,58]. As we previously reported, knockdown of SF2/SRSF1 in animals that overexpress TDP-43 in either motor neurons or all glia is sufficient to largely prevent the systemic effects on locomotion or lifespan, respectively [49]. Findings reported here knockdown of SF2/SRSF1 in either PNG or ALG partially rescued effects of TDP-43 pathology on lifespan (Fig 5), but importantly, this also extended cellular survival of the glial cells. This is in contrast with the effects of expressing caspase-3 inhibitors in glial cells that over-express TDP-43[42].

Like knockdown of SF2/SRSF1, inhibition of apoptotic cell death prevents glial cell death, but this exacerbates neuronal cell death and further shortens lifespan [37], the opposite sign of effect seen with SF2/SRSF1 knockdown (Fig 5). We therefore hypothesize that *SF2* decreases both the cell autonomous and non-autonomous toxicity of TDP-43 over-expression, either by interacting directly with TDP-43 or through some other mechanism.

## Methods

### Drosophila strains

Flies were reared at 21˚C and maintained on standard propionic acid food for all experiments in this study. A list of transgenic lines used in this study can be found in the table below. All were backcrossed to Canton-S derivative w$^{1118}$ (isoCJ1), our laboratory wild-type strain, for at least five generations. Female flies were utilized throughout the study. Strains used are shown in Table 2.

### Lifespan analysis

All flies were reared at 21˚C and shifted to 29˚C upon eclosion, in order to relieve the Gal80$^{ts}$ repression, allowing Gal4 mediated expression. Each lifespan experiments included 20 female flies per vial of a single genotype, with at least 60 flies total per genotype in each experiment. Flies were counted and flipped onto fresh food every other day. The Log-Rank (Mantel-Cox) test and the Gehan-Breslow-Wilcoxon test were used to compare the survival curves.

Genotypes used in Figs 1–5 are delineated in Table 3.

### Immunohistochemistry

Brains from adult drosophila were dissected in phosphate-buffered-saline (PBS) on days 2, 5, and 10. After dissection flies were transferred to 4% paraformaldehyde (PFA) solution (1XPBS, 4% PFA, and 0.2% Triton-X-100) and fixed for one hour in a vacuum to remove air from trachea. Brains were then washed 3 times in 1XPBST for 10 minutes each prior to either (1) being mounted directly for imaging; or (2) transferred to a blocking solution (10% normal

**Table 2. *Drosophila* Strains *used*.**

| |
|---|
| Canton-S derivative w$^{1118}$ (*isoCJ1*) |
| R85G01-Gal4 (PNG) |
| R54C07-Gal4 (SPG) |
| R54H02-Gal4 (CG) |
| R86E01-Gal4 (ALG) |
| R56F03-Gal4 (EG) |
| tub-Gal80$^{ts}$ (2$^{nd}$ chromosome) |
| 5X-UAS-unc84-2X-GFP (INTACT) |
| UAS-hTDP-43 (3$^{rd}$ chromosome) |
| UAS-hTDP-43 (2$^{nd}$ chromosome) |
| UAS-rpr |
| UAS-WM |
| UAS-SF2-IR |
| UAS-mcherry-IR |
| Repo-Gal4 |
| UAS-nlsGFP |
| tub-Gal80$^{ts}$ (3$^{rd}$ chromosome) |

**Table 3. Genotypes used in each figure.**

| Figure | *Drosophila* Strains |
|---|---|
| Figs 1B; 2A, B; 4A, 4B | *Tubulin-Gal80$^{ts}$/5XUAS-Unc84-2XGFP; UAS-TDP-43/ R85G01-Gal4* |
| | *Tubulin-Gal80$^{ts}$/5XUAS-Unc84-2XGFP; R85G01-Gal4/+* |
| Figs 1C; 2C, 2D; 4C, 4D | *Tubulin-Gal80$^{ts}$/5XUAS-Unc84-2XGFP; UAS-TDP-43/ R54C07-Gal4* |
| | *Tubulin-Gal80$^{ts}$/5XUAS-Unc84-2XGFP; R54C07-Gal4/+* |
| Figs 1D; 2E, 2F; 4E, 4F | *Tubulin-Gal80$^{ts}$/5XUAS-Unc84-2XGFP; UAS-TDP-43/ R54H02-Gal4* |
| | *Tubulin-Gal80$^{ts}$/5XUAS-Unc84-2XGFP; R54H02-Gal4/+* |
| Figs 1E; 2G, 2H; 4G, 4H | *Tubulin-Gal80$^{ts}$/5XUAS-Unc84-2XGFP; UAS-TDP-43/ R86E01-Gal4* |
| | *Tubulin-Gal80$^{ts}$/5XUAS-Unc84-2XGFP; R86E01-Gal4/+* |
| Figs 1F; 2I, 2J; 4I, 4J | *Tubulin-Gal80$^{ts}$/5XUAS-Unc84-2XGFP; UAS-TDP-43/ R56F03-Gal4* |
| | *Tubulin-Gal80$^{ts}$/5XUAS-Unc84-2XGFP; R56F03-Gal4/+* |
| Fig 1G | *Tubulin-Gal80$^{ts}$/5XUAS-Unc84-2XGFP; UAS-TDP-43/ Repo-Gal4* |
| Fig 3A–3C | *Tubulin-Gal80$^{ts}$/UAS-TDP-43; UAS-WM/ R85G01-Gal4* |
| | *Tubulin-Gal80$^{ts}$/UAS-rpr; UAS-WM/ R85G01-Gal4* |
| Fig 5A, 5D, 5E | *Tubulin-Gal80$^{ts}$/ UAS-SF2-IR; UAS-TDP-43/ Repo-Gal4* |
| | *Tubulin-Gal80$^{ts}$/ UAS-mcherry-IR; UAS-TDP-43/ Repo-Gal4* |
| Fig 5B | *Tubulin-Gal80$^{ts}$/ UAS-SF2-IR; UAS-TDP-43/ R85G01-Gal4* |
| | *Tubulin-Gal80$^{ts}$/ UAS-mcherry-IR; UAS-TDP-43/ R85G01-Gal4* |
| Fig 5C | *Tubulin-Gal80$^{ts}$/ UAS-SF2-IR; UAS-TDP-43/ R86E01-Gal4* |
| | *Tubulin-Gal80$^{ts}$/ UAS-mcherry-IR; UAS-TDP-43/ R86E01-Gal4* |

goat serum in 1X PBST; ThermoFisher: 31872) overnight at 4˚C on a nutator in preparation for staining. After blocking brains were washed once in 1X PBST for 10 minutes and transferred to primary antibodies against Repo (1:10, Developmental Studies Hybridoma Bank 8D12) and Elav (1:10, Developmental Studies Hybridoma Bank 7E8A10) in 1XPBST and incubated overnight at 4˚C on a nutator. Following primary antibody incubation brains were washed 4 times for 15 minutes each in a 3% salt solution (in 1X PBST) and transferred to secondary antibodies (Abcam: ab175673; ThermoFisher: A-21236) in 1XPBST and incubated overnight at 4˚C on a nutator. After secondary incubation brains were washed 4 times for 15 minutes each in a 3% salt solution (in 1X PBST) and mounted in FocusClear (CelExplorer).

## Confocal imaging and quantification

Brains were imaged within 24 hours of mounting using a Zeiss LSM 800 confocal microscope at 20X. All images were processed by Zeiss Zen software package. A single standard anterior section, defined by anatomy, was chosen for analysis for all cell quantification. Images were analyzed using FIJI (ImageJ). Mixed-effect analysis and two-way ANOVA were used to compare cell counts.

## TAPIN

Methods are described in detail in prior studies [50,51]. Briefly, 2 day old female flies were anesthetized with $CO_2$ and flash frozen in liquid nitrogen before being stored at -80˚C until an appropriate number of flies were collected for each group (>1,000,000 cells for all groups except SPG, for which cell number was >300,000). Fly heads were separated, homogenized, and incubated with an antibody against GFP (ThermoFisher: G10362) before being bound to protein A beads, cleaved, and rebound to protein G beads. Twice purified TAPIN-tagged nuclei were then burst and nuclear RNA was collected and stored at -80˚C.

### Library preparation and sequencing

TAPIN-generated nuclear RNA was treated with DNase and purified by the Arcturus PicoPure system. Purified RNA was amplified to cDNA via Nugen Ovation v2 system (Nugen: 7102–32) and was then fragmented with a Covaris s-series to 200bp. Fragmented cDNA was repaired and underwent second strand synthesis before being ligated to barcoded linkers (Nugen: 0319–32, 0320–32). Libraries were quantified via KAPA qPCR before being sequenced on Illumina NextSeq to 76bp read length.

### Computational analysis of RNA-seq libraries

Reads were aligned to the dm6 genome using STAR v2.7.6a [73], allowing for 4% mismatch and up to 100 alignments per read to ensure capture of young transposon sequences. Abundance of gene (UCSC curated Refseq, March 2018) and transposon (UCSC RepeatMasker) sequences was calculated with TEtranscripts v2.2.1 [74]. Differential analysis was performed using DESeq2 v1.28.1[98] using the DESeq normalization strategy and negative binomial modeling. B-H corrected FDR P-value threshold of $p < 0.05$ was used to determine significance. For correlation plots, Pearson correlation was calculated using normalized count values from DESeq2. For heatmap visualization, counts were normalized using a variance stabilizing transformation in DESeq2

### Statistical analysis

With the exception of the TAPINseq expression analysis, all statistical analyses were performed in GraphPad Prism v9.2. Specific statistical analyses are listed under each methods description and include mixed-effect analysis, two-way ANOVA, Log-rank (Mantel-Cox) test, and Gehan-Breslow-Wilcoxon test.

## Supporting information

**S1 Fig. Nuclear RNA-seq Schematic.** Flies expressing UAS-TDP-43 and the INTACT reporter (5XUAS-unc84-2XGFP) under control of a temperatures sensitive Gal80 plus a glial cell-type-specific Gal4 (methods) were (A) reared at 21˚C and shifted to 29˚C upon eclosion (day 0). Heads were collected for sequencing on day 2 and stored at -80˚C until use. (B) schematic of the INTACT nuclear tag used to purify nuclei and schematic of the tandem affinity purification process.
(PDF)

**S2 Fig. Gene expression profiles; overlap, PCA, and correlation plots.** Gene expression profiles generated from (A) Pearson correlation analysis demonstrating EG and CG more closely resemble each other at baseline than they do their own cell type with TDP-43 expression (B) principle component analysis plot demonstrating PNG and SPG separate from other glial cell types. (C) upregulated and downregulated genes ($p < .05$) in a given glial cell type and their overlap with other glial cell types.
(PDF)

**S3 Fig. Pathological expression of human TDP-43 in PNG causes loss of neurons.** The PNG-Gal4 and Gal80ts were used to induce post development expression of hTDP-43 and UAS- Watermelon (WM), a dual reporter module that encodes both a myristylated GFP that tethers to the membrane (myr-GFP-V5) and nuclear localized mCherry (H2B-mCherry-HA). As in other figures, flies were reared at 21˚C and shifted to 29˚C upon eclosion (day 0). To quantify effects on neuron number, brains were dissected at day 10 (D10) post temperature

shift. PNG glia were visualized using the mCherry nuclear reporter, and neuronal nulei were visualized using the Elav antibody. The numbers of Elav-labeled neuronal nuclei were quantified in A 4.68 μm stack (10 sections, 0.52 μm intervals) of ventral, posterior brain region as depicted with a yellow box in the image at left. This reveals a statistically significant (N = 15; Student's T-test; P = 0.0319) reduction in numbers of neurons in TDP-43 expressing (PNG-Gal4ts:UAS-TDP-43/UAS-WM) vs controls (PNG-Gal4ts>UAS-WM). Means and SEM are shown.
(PDF)

**S1 Table. Numbers of fly heads used for TAPIN purification and sequencing based on number of glial cells of each type per brain.**
(PDF)

**S2 Table. Statistically significant differentially expressed features.**
(XLSX)

**S3 Table. Statistically significant differentially expressed TEs.**
(XLSX)

**S4 Table. DNA damage/repair pathways.**
(XLSX)

**S5 Table. Chromatin organization pathway.**
(XLSX)

**S6 Table. Nucleocytoplasmic transport pathway.**
(XLSX)

**S7 Table. Inflammatory response pathways.**
(XLSX)

**S8 Table. SF2 expression across glial cell types. SF2 expression was significantly reduced at day 2 post induction of TDP-43 in SPG and PNG day 2.**
(PDF)

## Acknowledgments

We are grateful to Roger Sher, Lillian Talbot, Shreevidya Korada, Enas Gad El-Karim, Richard Keegan, Jorge Azpurua, Meng-Fu Shih, and Yung-Heng Chang for many helpful discussions. We also thank Yung-Heng Chang for comments on the manuscript, and Gilbert L Henry for technical help and advice with the nuclear sequencing methods.

## Author Contributions

**Conceptualization:** Sarah Krupp, Josh Dubnau.

**Data curation:** Oliver Tam, Gale M. Hammell.

**Formal analysis:** Oliver Tam.

**Funding acquisition:** Josh Dubnau.

**Investigation:** Sarah Krupp, Isabel Hubbard.

**Methodology:** Sarah Krupp.

**Project administration:** Josh Dubnau.

**Resources:** Gale M. Hammell, Josh Dubnau.

**Supervision:** Gale M. Hammell, Josh Dubnau.

**Visualization:** Sarah Krupp, Oliver Tam.

**Writing – original draft:** Sarah Krupp.

**Writing – review & editing:** Josh Dubnau.

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
