## [Decision Letter · Decision Letter 0]

29 May 2023

Dear Dr Dubnau,

Thank you very much for submitting your Research Article entitled 'TDP-43 pathology in Drosophila induces glial-cell type specific toxicity that can be ameliorated by knock-down of SF2/SRSF1.' to PLOS Genetics.

The manuscript was fully evaluated at the editorial level and by independent peer reviewers. The reviewers appreciated the attention to an important topic but identified some concerns that we ask you address in a revised manuscript.

We therefore ask you to modify the manuscript according to the review recommendations. Your revisions should address the specific points made by each reviewer.

Yours sincerely,

Fengwei Yu

Academic Editor

PLOS Genetics

Gregory P. Copenhaver

Editor-in-Chief

PLOS Genetics

Reviewer's Responses to Questions

**Comments to the Authors:**

Reviewer #1: Krupp et al. described the effect of SF2/SRSF1 knockdown in glia-induced TDP-43 pathology in Drosophila. Overall the study is interesting, of interest, and well conducted. I have some major concerns prior to publication:

1. How physiologically relevant is glial-specific TDP-43 overexpression? TDP-43 accumulation occurs in both glia and neurons, but is TDP-43 expressed in both cell types? If it’s dominantly expressed in neurons, overexpressing TDP-43 in glia does not recapitulate the situation under normal physiological condition.

2. For all experiments in the study, does the control have similar GAL4:UAS ratio as the experimental group? As titration effect of gene expression might occur when using one GAL4 for two UAS, it will be best to compare all effects with similar number of UAS transgenes for the same GAL4.

3. Did the authors try overexpressing SF2/SRSF1 in the presence of TDP-43 in glia? It is a bit counter-intuitive to learn that knockdown of SF2/SRSF1 ameliorate TDP-43 pathology in glia, as RNA-sequencing results suggest that this gene is downregulated when TDP-43 is overexpressed. One would imagine that by overexpressing SF2/SRSF1 TDP-43 pathology will be rescued.

4. The discussion on the non-cell-autonomous toxicity from glia to neurons is a bit hard to understand. It will help if the author can revise and add in more discussion to make things more clear.

5. There is a typo in Figure 1A, the genotype is unc8402xGFP?

Reviewer #2: Understanding how TDP-43 related pathology and aggregation can underlie neurodegenerative diseases like ALS, FTD, and AD is a critical current goal for the field. Important recent work highlighted that not only neurons play a role in the pathological process, but that non-cell autonomous interactions with glia are important. Krupp and colleagues examine such interactions in more depth in this manuscript, to determine the effects of TDP-43 overexpression in specific subpopulations of glia on TDP-43 pathology. They show that organismal survival is most severely impacted following TDP-43 overexpression in perineurial glia or astrocytes; for perineurial glia, this is not due to cell ablation, but likely non-cell autonomous defects. Further, they observe that TDP-43 overexpression in each glial cell type leads to varying degrees of age-dependent cell loss. The authors conduct TAPIN-seq to determine how TDP-43 overexpression affects the transcriptome of each subpopulation of glial cells and show that SF2/SRSF1 is reduced in perineurial glia and astrocytes, consistent with previous broad cell-type findings in ALS models. Further knockdown of SF2/SRSF1 in either perineurial glia or astrocytes reduces TDP-43 pathology including cell loss and lifespan.

This paper represents a strong and important contribution to the field for two main reasons: 1) it demonstrates a fascinating biological finding that glial subtypes are not equally susceptible / causative in TDP-43 pathology and 2) it establishes a valuable resource in the nuclear RNA sequencing databases of different glial subtypes under normal conditions and TDP-43 overexpression conditions. Both aspects are notable advances in the field. Moreover, the paper is well-written, with the experimental logic clearly laid out and the results straightforward and easy to follow. My criticisms are minor and largely restricted to issues of data presentation and the need for additional discussion on the roles of glial subtypes and the mechanism of SF2/SRSF1 function. With these minor additions, the paper will be suitable for publication.

Experimental Point - Though consistent with data from the field, the knockdown of SF2/SRSF1 that rescues TDP-43 pathology when TDP-43 pathology already results in a reduction of SF2/SRSF1 is counterintuitive. If the two were causative / genetically linked, I expect SF2/SRSF1 overexpression would rescue the pathology. Can you instead overexpress SF2/SRSF1 and see altered TDP-43 pathology? Further, the authors posit that SF2/SRSF1 “rescue” results in “a reduction in the non-cell autonomous toxicity of the glia to surrounding neurons.” Can this be demonstrated and quantified? It would help strengthen the existing data. It would also be helpful for the authors to include further discussion of the role / mechanism of SF2/SRSF1.

Discussion Point - There is little discussion of the role that astrocytes are playing in TDP-43 overexpression pathology. The roles of subperineurial glia and perineurial glia are well discussed, but additional discussion / speculation on the role of astrocytes would be very helpful here.

Data Presentation Points - My biggest issue with this paper is in data presentation. In its current state, it is very challenging to discern labels and interpret the results of figures and graphs throughout the paper. Though the issue is present in all figures, it is most distinctly so with Figure 2. For all figures, can the authors adjust font size and clarity? In Figure 2, even at very high magnification, it’s still challenging to tell what the labels are showing. Additionally, larger graphs (i.e., 2B, D, F, H, J) would be very helpful in conveying the important points as with the rest of the figures. In all figures, larger image labels would be helpful.

In Figure 1, it is also difficult to tell the difference in the curve genotypes due to the size of the shapes. Can you color the TDP-43 containing curves so it’s easier to see? Also, can you denote significance on the graph? There is no significance in the figure despite noting it in the text and a p value in the figure legend.

Finally, in Figure 4, the letter labeling is inconsistent with the rest of the paper. Also, the pink bar denoting the Tg_TDP43 genotype is difficult to see. Can it be changed to aid in visibility?

Reviewer #3: TDP43 pathology plays a central role in the pathogenesis of ALS and FTD, and may be also involved in AD as well. Thus, it has been under intense investigation in the field. Over the years, Dr. Josh Dubnau and his colleagues have investigated the effects of TDP43 pathology in Drosophila by overexpressing it in different cell types and have made a series of important discoveries (such as DNA damage and retrotransposon activation). In the current study, the authors made an interesting observation that TDP43 expression in the perineural glia (PNG) and astrocytes had most pronounced effect on organismal survival compared with other types of glia. Because genetic ablation of PNG did not have the same effect, these findings suggest that some unknown factors released by TDP43-expressing PNG cause systematic toxicity and neurodegeneration. This is important, because they are now in an excellent position to perform genetic studies to identify the nature of these factors released by PNG. Of course, their work here would be much more significant and novel if they could report what these factors are. Then they went on to perform cell-type specific nuclear RNA-seq and found that SF2/SRSF1 is downregulated in PNG expressing TDP-43. Further KD of SF2/SRSF1 in PNG rescued loss of these glia and reduced systematic toxicity of TDP43. Although the beneficial effects of SF2/SRSF1 KD in TDP-43 or C9 models are already known, their RNA-seq datasets will be useful for the field.

The authors can consider the following suggestions for minor revisions to further improve their manuscript.

1. The strength of different glial subtype-specific Gal4 drivers may be different. Is it possible different levels of TDP43 are expressed in different glial subtypes to cause different effects on organismal survival? Western blot analysis will not help address this question because the numbers and location of glia in different subtypes are different. Or immunostaining of different flies is good enough to tell the difference. At a minimal, the authors can discuss more in detail the number and location of glial subtypes, and potentially differential strengths of different Gal4 drivers.

2. As mentioned above, SF2/SRSF1 is already known to be a modifier in TDP43 and C9 models (e.g., PLoS Genet. 2021). If not too much work, the authors can include another novel modifier identified from their RNA-seq analysis to enhance the novelty of the study. Alternatively, they can perform a few more mechanistic studies to understand further how SF2/SRSF1 modifies TDP-43 toxicity. For instance, is SF2/SRSF1 required to export TDP43 mRNA from the nucleus to the cytoplasm as it does for C9 repeats as reported? Does SF2/SRSF1 overexpression enhance the phenotype? And so on.

3. Maybe the authors published this in their earlier papers. In several places, the authors talk about toxicity of glial TDP43 on neurons, but they never showed any data. Is that possible to include neuron survival data at least in one figure?

4. In Figure 1 legend, it is stated that p<0.0001 for all panels. Is that true? This reviewer is a little bit surprised, because in Panel C, the small difference is largely due to the second data point and death occurred only few days after the experiment started. So the authors should carefully perform statistical analyses again and state in the legend what statistical methods were used. The number of flies for each panel should be stated and it is better to list p values in each panel of the figure as well.

5. In Figure 2 and Figure 3 legends, *p<0.5 must be an error. Is it *p<0.05? Also, please state the values are S.D. or S.E.M., what are n numbers, and by what statistical analysis in all figure legends.

6. In Figure 3B, the number of PNG cells at day 2 is >200, but the image does not show that many cells as in Figure 2. Please explain.

7. In Figure 4 legend, they say “upregulated (blue) and downregulated (red) genes”. But in the figure, upregulated genes are in red and downregulated genes are in blue.

8. Figure 5 can be reorganized to better utilize the empty spaces. Panels D and E can be new Panels A and B, and Panels A-C can be new Panels C-E.

9. Please increase font size in all figures.

**Have all data underlying the figures and results presented in the manuscript been provided?**

Reviewer #1: Yes

Reviewer #2: Yes

Reviewer #3: None

PLOS authors have the option to publish the peer review history of their article (what does this mean?). If published, this will include your full peer review and any attached files.

Reviewer #1: No

Reviewer #2: No

Reviewer #3: No

---

## [Decision Letter · Decision Letter 1]

13 Sep 2023

Dear Dr Dubnau,

We are pleased to inform you that your manuscript entitled "TDP-43 pathology in Drosophila induces glial-cell type specific toxicity that can be ameliorated by knock-down of SF2/SRSF1." has been editorially accepted for publication in PLOS Genetics. Congratulations!

Yours sincerely,

Fengwei Yu

Academic Editor

PLOS Genetics

Gregory P. Copenhaver

Editor-in-Chief

PLOS Genetics

Comments from the reviewers (if applicable):

Reviewer's Responses to Questions

**Comments to the Authors:**

Reviewer #1: The reviewer has no further questions and the manuscript is ready for publication in its current form

Reviewer #2: I thank the authors for their thoughtful replies to my comments, for the addition of new data in Figure S3, and especially the changes to data presentation which greatly improved the figures. This was very helpful. The authors have appropriately addressed all of my comments and I can support publication. Thank you!

Though I'd like to cast my vote (since all 3 reviewers asked for this) for overexpressing SF2/SRSF1 in the next paper!

Reviewer #3: The authors have satisfactorily addressed most of my comments. Some suggested experiments were not performed, such as SF2/SRSF1 overexpression, which was also proposed by another reviewer. I understand their reasoning why it is more important to focus on down-regulation.

**Have all data underlying the figures and results presented in the manuscript been provided?**

Reviewer #1: None

Reviewer #2: Yes

Reviewer #3: Yes

PLOS authors have the option to publish the peer review history of their article (what does this mean?). If published, this will include your full peer review and any attached files.

Reviewer #1: No

Reviewer #2: No

Reviewer #3: **Yes: **Fen-Biao Gao

**Data Deposition**

http://datadryad.org/submit?journalID=pgenetics&manu=PGENETICS-D-23-00514R1

**Press Queries**

---

## [Editor Report · Acceptance letter]

20 Sep 2023

PGENETICS-D-23-00514R1 

TDP-43 pathology in Drosophila induces glial-cell type specific toxicity that can be ameliorated by knock-down of SF2/SRSF1. 

Dear Dr Dubnau, 

We are pleased to inform you that your manuscript entitled "TDP-43 pathology in Drosophila induces glial-cell type specific toxicity that can be ameliorated by knock-down of SF2/SRSF1." has been formally accepted for publication in PLOS Genetics! Your manuscript is now with our production department and you will be notified of the publication date in due course.

With kind regards,

Zsofi Zombor

PLOS Genetics

On behalf of:
